# Cortisol Awakening Reaction and Anxiety in Depressed Coronary Artery Disease Patients

**DOI:** 10.3390/jcm11020374

**Published:** 2022-01-13

**Authors:** Cora Weber, Stella V. Fangauf, Matthias Michal, Joram Ronel, Christoph Herrmann-Lingen, Karl-Heinz Ladwig, Manfred Beutel, Christian Albus, Wolfgang Söllner, Frank Holger Perschel, Martina de Zwaan, Kurt Fritzsche, Hans-Christian Deter

**Affiliations:** 1Department of Psychosomatic Medicine and Psychotherapy, Clinic Hennigsdorf, Oberhavel Clinic, 16761 Hennigsdorf, Germany; 2Department of Psychosomatics and Psychotherapy, Campus Benjamin Franklin, Charité Universitätsmedizin Berlin, 12203 Berlin, Germany; deter@charite.de; 3Department of Psychosomatic Medicine and Psychotherapy, University of Göttingen Medical Center, 37075 Göttingen, Germany; sfangauf@outlook.com (S.V.F.); cherrma@gwdg.de (C.H.-L.); 4Department of Psychosomatic Medicine and Psychotherapy, University Medical Center Mainz, 55131 Mainz, Germany; matthias.michal@unimedizin-mainz.de (M.M.); manfred.beutel@unimedizin-mainz.de (M.B.); 5Department of Psychosomatic Medicine, Klinik Barmelweid, 5017 Barmelweid, Switzerland; Joram.Ronel@barmelweid.ch; 6Department of Psychosomatic Medicine and Psychotherapy, University Hospital Rechts der Isar, Technical University of Munich, 81675 Munich, Germany; karl-heinz.ladwig@tum.de; 7Department of Psychosomatics and Psychotherapy, University of Cologne, 50937 Cologne, Germany; christian.albus@uk-koeln.de; 8Department of Psychosomatic Medicine and Psychotherapy, Paracelsus Medical University, Nuremberg Medical Hospital, 90419 Nuremberg, Germany; Wolfgang.Soellner@klinikum-nuernberg.de; 9Institute of Laboratory Medicine, Clinical Chemistry and Pathobiochemistry, Charité Universitätsmedizin Berlin, 13353 Berlin, Germany; frank.perschel@charite.de; 10Labor Berlin, Charité Vivantes GmbH, 13353 Berlin, Germany; 11Department of Psychosomatic Medicine and Psychotherapy, Hannover Medical School, 30625 Hannover, Germany; deZwaan.Martina@mh-hannover.de; 12Department of Psychosomatic Medicine and Psychotherapy, University Medical Center Freiburg, 79104 Freiburg, Germany; kurt.fritzsche@uniklinik-freiburg.de

**Keywords:** coronary artery disease, depression, anxiety, cortisol, HPA axis, cortisol awakening reaction, area under the curve with respect to ground, area under the curve with respect to increase, psychosocial stress

## Abstract

Disturbances of HPA axis functioning as represented by cortisol awakening reaction (CAR) belong to the mediating pathways linking psychosocial distress and cardiovascular risk. Both depression and anxiety have been confirmed as independent risk factors for coronary artery disease (CAD). However, data on anxiety and cortisol output in CAD patients are scarce. Based on previous data, we hypothesized that anxiety would be associated with higher cortisol output and a more pronounced morning increase in moderately depressed CAD patients. 77 patients (60 y, 79% male) underwent saliva sampling (+0, +30, +45, +60 min after awakening, midday and late-night sample). Anxiety was measured using the Hospital Anxiety and Depression Scale (HADS) and patients were grouped into anxious versus non anxious subjects based upon the recommended score (≥11). A repeated measures ANOVA yielded a significant time and quadratic time effect referring to the typical CAR. Anxious patients showed a significantly steeper 30 min increase, higher AUCi, lower waking and late-night cortisol levels. The steeper cortisol increase in the anxious group is in line with previous data and may be interpreted as a biological substrate of affect regulation. The lower basal and late-night levels coupled with greater AUCi mirror a more dynamic reactivity pattern compared to depressed subjects without anxiety.

## 1. Introduction

Coronary artery disease (CAD) represents a major cause of morbidity and mortality worldwide [1]. Both depression [2,3,4] and anxiety [5,6] have been posited as independent risk factors for both incidence and prognosis of CAD. Altered functioning of the hypothalamic pituitary adrenal (HPA) axis are thought to be among the mediating pathways involved in CAD as well as in depression and anxiety [7,8], promoting hypertension, procoagulant activity, and other cardiac risk factors [9,10,11].

Data on cortisol and depression have brought equivocal results with both increased and decreased levels being reported [12]. In CAD patients, flattened cortisol slopes over the day were found in depressed individuals [13,14]. Waller et al. [14] reported blunted cortisol reactivity to a psychosocial stress test (Trier Social Stress Test, TSST) in depressed CAD but not in depressed non-CAD patients.

Since the underlying atherosclerosis is conceived as a process of chronic subclinical inflammation, a decreased HPA activity level might contribute to a worse prognosis in depressed CAD subjects. The phenomenon of blunted HPA activity as a feature of chronic stress and posttraumatic stress disorder reverses the traditional paradigm of hypercortisolism as a natural endocrine reaction to stress going back to the early work of Selye [15]. Deficient cortisol signaling may contribute to disease progression.

There are fewer studies on anxiety and HPA axis functioning in non-CAD populations, reporting inconsistent findings, and even fewer studies in CAD patients [16]. A current metaanalysis did not reveal significant differences between patients with and without anxiety disorders in the response to psychosocial stress neither in AUGi (area under the curve with respect to increase) nor in AUCg (area under the curve with respect to ground) [17].

In a study by Mantella et al., older adult (*n* = 111; mean age 74 years) subjects with generalized anxiety disorder (GAD) versus no GAD showed higher cortisol levels and a higher cortisol output over the day [18]. Accordingly, Vreeburg and colleagues found higher saliva cortisol levels (+0, +30, +45, +60 min after awakening) in patients with anxiety disorder [19]. In contrast, in a study by Heaney and colleagues, only in younger adults a positive association of area under the curve (AUC) and cortisol awakening reaction (CAR) with anxiety was found, whereas in older adults this association was not present [20]. In a Dutch population-based study in 1788 subjects aged 65 and older by Hek et al., adults with an anxiety disorder (*n* = 145) showed a lower CAR (saliva sampling at +0 and +30 min) [21]. In sum, data on anxiety and cortisol show equivocal results. In non-depressed CAD patients, a previous study from our group revealed a higher AUCg and a trend for higher AUCi in anxious versus non-anxious subjects [16].

Based on this previous study and on some of the literature, we hypothesized a greater CAR in a sample of moderately depressed CAD patients grouped by anxiety. We expected a steeper increase (AUCi) and a higher total cortisol output (AUCg) in the anxious subjects. Secondly, we aimed to explore associations with somatic (age, BMI, LVEF) and psychosocial factors (vital exhaustion (VE), quality of life (QOL), coping with illness). 

## 2. Materials and Methods

### 2.1. Study Design

The study sample was taken from a large randomized controlled multicenter trial comparing usual care plus a stepwise psychotherapy intervention to usual care plus one individual information session (SPIRR-CAD, NCT00705965). The design and main results have been described in more detail elsewhere [22,23]. The trial took place in Germany and enrolled 570 patients with stable CAD and mild to moderate depression (Hospital Anxiety and Depression Scale, HADS ≥ 8, [24]). No difference was found in the intervention versus the control group with regard to depression decrease. However, Type D patients in the intervention group showed a greater reduction of depression referring to a differential benefit from the stepwise psychotherapy intervention [23].

The present sub study assessed associations of cortisol with psychosocial and somatic measures. Three of 10 study centers took part in data collection (Berlin, Mainz, and Munich, Germany). All ethics committees of the study sites approved the trial protocol. The study was conducted in accordance with good clinical practice and the Helsinki Declaration. All patients gave written informed consent before being enrolled.

### 2.2. Participants

In the present secondary analysis, we analyzed data from 77 of the 570 patients with complete saliva cortisol samples. The following inclusion criteria were applied: angiographic or clinical evidence of CAD, depression score ≥ 8 on the HADS depression subscale, no severe heart failure (left ventricular ejection fraction (LVEF) <20% or New York Heart Association (NYHA) class IV), sufficient knowledge of German, no life-threatening mental or physical diseases. 

### 2.3. Assessments

#### 2.3.1. Cortisol Sampling Protocol and Analyses

Saliva samples were collected via salivettes (^®^Salivette, Sarstedt, Nümbrecht, Germany) following a standardized sampling protocol (+0, +30, +45, +60 min after individual awakening as well as a midday (between 12 pm and 3 pm) and a late-night probe taken at 30 min before each individual’s bedtime, but not later than midnight). Patients were instructed to place the first salivette directly beside their bed and instructed not to eat, not to smoke, and not to brush their teeth within the time span of saliva collection, that is, within the first hour after awakening, and before taking the late-night probe. They were only allowed to drink plain water. 

Recovery from the swap included in the collection kit was performed by centrifugation at 1000× *g* according to recommendation of the manufacturer of the kit. Recovered saliva samples were stored at −20 °C until analysis. Cortisol concentration was measured using a solid-phase 125I radioimmunoassay (Coat-A-Count® Cortisol RIA, DPC Biermann GmbH, Bad Nauheim, Germany) counted in a LKB Wallac 1277 Gammamaster Automatic Gamma Counter (Perkin Elmer, Rodgau, Germany). Due to expected low cortisol concentrations in the saliva samples, the following modifications were made in handling the RIA [25]: standards were prediluted 1:10 with distilled water, 200 μL of the diluted standards, and saliva samples were pipetted instead of 25 μL as recommended for serum samples. Incubation was carried out for 45 min at 37 °C. All samples were measured in duplicate.

#### 2.3.2. Medical History and Questionnaires

Medical histories were taken from the patients’ charts and standardized interviews. Anxiety was assessed by the German version of the Hospital Anxiety and Depression Scale (HADS; [26]). It is a two-scale instrument including seven items on anxiety and seven items on depression. To only include cases of high symptomatology, a cut-off of ≥11 was chosen to discriminate between anxious versus non-anxious patients, as recommended by Zigmond and Snaith [24]. Coincidentally, this clinical cut-off score conformed with the median of our sample. 

The primary study included the German versions of various validated questionnaires, including the 9-item Patient Health Questionnaire (PHQ-9), to assess depression [27,28], the Maastricht Questionnaire (MQ) for vital exhaustion (i.e., symptoms of feeling exhausted, rejected, and defeated; [29]), and the 5-scale Freiburg Questionnaire of Coping with Illness to examine the patients’ coping style [30]. To assess health-related quality of life, the Medical Outcomes Short Form Health Survey (SF-36) was used [31]. This 36-item questionnaire contains eight subscales, namely physical functioning, vitality, physical role functioning, bodily pain, general health perceptions, social role functioning, mental health, and emotional role functioning. Two summarized and z-standardized component scores can be calculated from the physical and mental health subscales, respectively. Additionally, the two components of type D personality, i.e., social inhibition and negative affectivity, were assessed using the 14-item Type D Scale (DS-14; [32]). 

### 2.4. Data Analysis

Data were analyzed using SPSS (Version 25, IBM Corp., Armonk, NY, USA). Data were tested for normality, and parametric and non-parametric tests were used appropriately. Significance was set at *p* < 0.05. AUCg and AUCi were computed using standardized formulas [7]. 

To characterize the sample, we checked for differences of AUCg and AUCi with regard to somatic and psychometric variables, by either the Mann Whitney-U-test or the ANOVA, as indicated. We conducted bivariate Pearson’s correlation analyses checking for associations of cortisol measures (AUCg and AUCi) with somatic and psychometric parameters.

We compared anxious versus non anxious patients with regard to clinical background variables by t-test or Chi squared test, as appropriate. To compare anxious versus non anxious patients with regard to the CAR, we calculated a repeated-measures ANOVA (four steps, i.e., 4 time points of the CAR: +0, +30, +45, +60 min; two groups, i.e., anxious versus non anxious subjects). The model was adjusted for possible confounders. In cases where Mauchly’s test of sphericity was significant, Greenhouse-Geisser corrected values are reported. 

## 3. Results

The majority of the 77 patients included were male (79%), with a mean age of 59.5 years (SD = 8.4). Most of them were on beta blockers, statins and ACE-inhibitors, and most had a history of myocardial infarction (MI). Indicating severity of CAD, the mean number of affected coronary vessels was 2.3 (SD = 1.1). More detailed characteristics are presented in Table 1. 

There were no significant differences in AUCg or AUCi with regard to NYHA class, hypertension, diabetes, history of MI, smoking, beta blockers, statins, antidepressants, marital status, socio-economic status (SES), employment status, nor Type D. Women compared to men showed significantly lower AUCg (t(38.57) = 2.27, *p* = 0.029). Furthermore, patients with versus without hyperlipidemia showed lower AUCg (t(75) = 2.43, *p* = 0.017). Comparing anxious versus non-anxious subjects with regard to somatic and psychometric variables (see Table 2), we found significant differences with regard to age (t(75) = 2.56, *p* = 0.013), anxious subjects being significantly younger, smoking status (χ² = 5.43, *p* = 0.02), having more smokers in the non-anxious group, type D personality (χ² = 6.68, *p* = 0.01), with more patients with type D in the anxious group, and SES (χ² = 6.22, *p* = 0.045), with higher SES in the anxious group.

The individual CAR measures only differed at +0 min with significantly lower levels for anxious versus non-anxious patients (t(50.24) = 2.56, *p* = 0.014) (see Figure 1).

Also, at 30 min before bedtime, cortisol levels were significantly lower in the anxious subjects ((t(34.50) = 3.12, *p* = 0.003).

### 3.1. Correlation Analyses

There were no significant associations of AUCg and AUCi with age, BMI, or extent of CAD as measured by the number of affected coronary vessels. However, AUCg was significantly negatively correlated with LVEF (r = −0.279, *p* = 0.032). There was also a non-significant trend for AUCi and anxiety (r = 0.218, *p* = 0.057). All other psychometric variables (depression scores as assessed with the HADS and PHQ-9, vital exhaustion (MQ), quality of life (SF-36), and coping with illness (FKV)) were not significantly associated with either AUCg or AUCi. Checking for an association of the extent of CAD as measured by the number of affected vessels with anxiety, we found no significant correlation. Correlation analyses are available as Appendix A (see Appendix A). 

### 3.2. Repeated-Measures Analyses

Repeated-measures ANOVA revealed a significant main effect of time (F(2.344, 175.782) = 9.003, *p* < 0.0001). We also found a significant interaction between time and group (F(2.344, 175.782) = 5.355, *p* = 0.003). There was also a significant quadratic time effect (F(1, 75) = 24.584, *p* < 0.0001) and a significant quadratic time by group interaction (F(1, 75) = 6.077, *p* = 0.016; see Figure 1). The anxious group showed a steeper increase within the first 30 min of awakening. Data can be seen in the Appendix A.

In a second step, we included potential confounders. First, we included LVEF and hyperlipidemia, as they were significantly associated with AUCg. Secondly, we included smoking status, Type D, and SES. Both analyses confirmed the significant main effect of time and the significant time by group interaction. Lastly, we added age and sex, which confirmed the significant time by group interaction. Data are available as Appendix A (Appendix A).

Comparing AUCg and AUCi between anxious and non-anxious subjects, a significant difference was detected for AUCi (U = 460.0, z = −2.738, *p* = 0.006), with anxious subjects showing a higher AUCi.

## 4. Discussion

In the present study, we saw a typical CAR pattern within the first 60 min of individual awakening, with a rapid increase of cortisol levels within the first 30 min, followed by a slower decrease, in a sample of moderately depressed patients with stable CAD. Cortisol concentrations at awakening and at bedtime were significantly lower in the anxious versus non-anxious subjects. However, the cortisol increase within the first 30 min of awakening was steeper in the anxious subjects, going along with higher AUCi. These effects were independent of all tested confounders.

In contrast to our second hypothesis and the previous study from our group [16], we did not see a higher total cortisol output within the first 60 min after awakening (AUCg) in the anxious subjects.

The finding of lower basal and late night cortisol in the anxious patients appears somewhat counterintuitive, since one may rather expect higher levels as a biological substrate of fear; however, associations seem to be more complex.

Thus, the reported evidence on the association of anxiety and HPA axis functioning is inconsistent, and in sum, studies are scarce compared to those on depression. While some authors reported higher cortisol levels in older adults with GAD [18], others found no association of cortisol levels with an anxiety symptom score [20]. Hek et al. [21] described lower cortisol levels at +30 min after awakening and at 5:00 p.m. coupled with a lower CAR in elderly adults with longstanding anxiety disorders, especially GAD, and attributed this to downregulation of HPA axis activity as a result of chronic anxiety disorder.

This latter study is in contrast to our findings of lower cortisol levels at awakening (+0) and at bedtime and a higher AUCi in moderately depressed anxious CAD patients. In fact, we did see a significant CAR regarding the whole sample, but splitting them into anxious and non-anxious, the anxious seemed to present a more dynamic CAR pattern with a significantly steeper increase in the first 30 min after awakening.

In contrast to the study by Hek et al., our patients were all moderately depressed. Conceiving of depression as a chronic stressor, one may interpret the difference between anxious and non-anxious subjects as an affect driven endocrine reaction which would accord with the previous literature [33,34].

Typically, with acute stress, the HPA axis is activated, resulting in elevated cortisol output. With the passage of time, under the influence of chronic stress, the body may mount a counter-regulatory response such that cortisol output rebounds below normal [33]. This would be a biologically plausible explanation, since the HPA axis is regulated by a potent negative feedback circuit [33]. Low cortisol output has been documented in chronically stressed non-psychiatric populations, individuals with somatoform disorders, fibromyalgia, chronic fatigue, and posttraumatic stress disorder [33,34]. A persistent lack of cortisol availability in chronically stressed individuals is thought to increase vulnerability for development of further stress-related diseases [34]. Thus, we might even conceive of anxiety as a protective factor in our moderately depressed CAD patients, with chronic depression conferring a heightened risk of further cardiovascular events as known [2,3].

In accordance with this, two previous studies reported flattened cortisol slopes in depressed CAD patients, both during the day [13], and in response to an acute psychosocial stressor [14]. The authors hypothesized that this blunted cortisol pattern may advance atherosclerosis by reduced suppression of inflammation. In the light of these hypotheses and the literature, it seems plausible that the steeper increase after awakening observed in our anxious moderately depressed CAD patients is indeed moderated by anxiety. The finding of higher AUCi is also backed by our previous study on anxious non-depressed CAD patients [16].

At the same time, the preselection of depressed CAD subjects in the present study may convey a floor effect with reduced selectivity of psychological markers, which would explain the lack of difference in AUCg between anxious and non-anxious subjects in contrast to Merswolken et al. [16]. In accordance with Miller and colleagues [33], anxious subjects may have more anxiety-associated cortisol output during the day with a following depletion of stores which is mirrored by lower late night and basal levels, in sum resulting in a more dynamic regulation compared to moderately depressed CAD patients without anxiety.

Some limitations to this study should be considered: (a) The present study was a secondary analysis from a larger trial [23]. Therefore, the sample was preselected including only patients with stable CAD and moderate levels of depression. Floor effects and reduced selectivity may mask stronger effects; (b) Cortisol levels were available for a subgroup of 77 patients, with a majority being male. Findings need replication in larger samples, and especially gender differences as suggested should be checked in larger samples; (c) A further limitation is the one-day sampling protocol at home without external supervision. Therefore, the results have to be interpreted with caution.

## 5. Conclusions

Anxious subjects show a more dynamic cortisol pattern including lower waking and late-night cortisol levels coupled with a steeper 30 min increase compared with non-anxious subjects in a sample of moderately depressed CAD patients. Chronic stress such as depression may result in flatter cortisol rhythms and contribute to the progression of coronary atherosclerosis as described before. Thus, anxiety may represent a protective factor in a potentially life threatening disease such as CAD, but the complex field of anxiety in CAD needs further study, preferably including novel biological data which our group has been working on in other contexts [35,36]. In sum, the findings need to be interpreted with caution due to the methodological limitations described above.

## Figures and Tables

**Figure 1 jcm-11-00374-f001:**
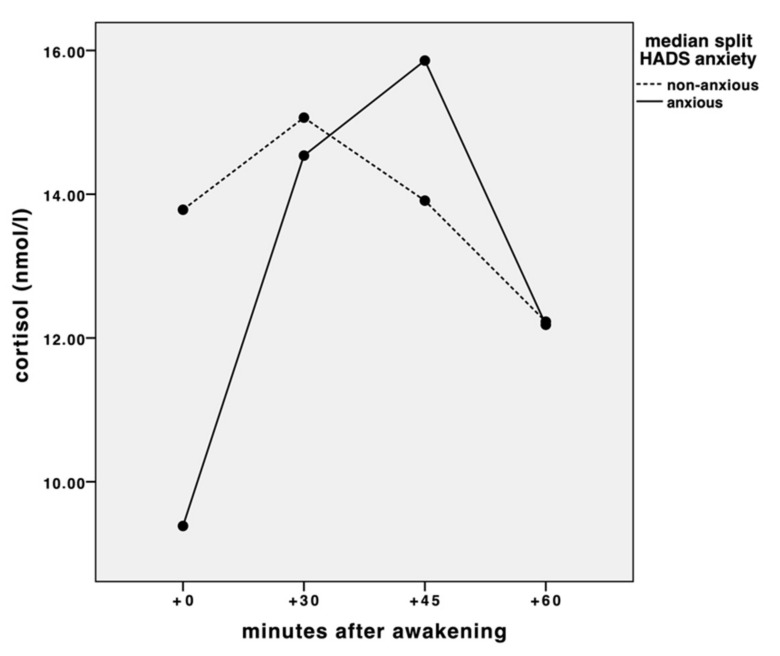
Cortisol awakening response (CAR) in anxious versus non anxious subjects (+0, +30, +45, +60 min after awakening).

**Table 1 jcm-11-00374-t001:** Baseline Characteristics.

	*n*	%
Male sex	61	79.2
Married	48	62.3
Socioeconomic status		
Low	25	32.5
Medium	33	42.9
High	19	24.7
Employed	28	36.4
NYHA class		
I	27	35.1
II	40	51.9
III	10	13.0
Hyperlipidemia	61	79.2
Hypertension	66	85.7
Diabetes mellitus	19	24.7
History of MI	44	57.1
Smoking	22	28.6
Beta-blocker	69	89.6
Statins	73	94.8
ACE-inhibitors	45	58.4
Antidepressants	18	23.4
Type D	56	72.7
	**M**	**SD**
Age, y	59.5	8.4
LVEF	63.5	13.3
BMI	29.3	5.6
CCI	2.3	1.6
Aerobic exercise (min/week)	374.7	257.4
Number of affected coronary vessels	2.2	1.1
Cortisol (nmol/L) +0	11.3	7.3
Cortisol (nmol/L) +30	14.8	6.5
Cortisol (nmol/L) +45	15.0	6.5
Cortisol (nmol/L) +60	12.2	5.8
HADS depression	10.5	2.6
HADS anxiety	11.2	3.5
PHQ depression	10.3	4.3
DS-14 negative affectivity	16.2	4.3
DS-14 social inhibition	12.4	5.3
MQ vital exhaustion	27.2	10.0
SF-36 physical health sum score	36.8	10.2
SF-36 mental health sum score	37.6	10.1

Abbreviations: BMI = body mass index (kg/m²), CCI = Charlson comorbidity index, DS-14 = fourteen item Type D scale, HADS = Hospital Anxiety and Depression Scale, IQR = interquartile range; LVEF = left ventricular ejection fraction, M = mean, MQ = Maastricht Vital Exhaustion Questionnaire, NYHA = New York Heart Association, PHQ = Patient Health Questionnaire, SF-36 = 36-item Medical Outcomes Short Form Health Survey, SD = standard deviation.

**Table 2 jcm-11-00374-t002:** Baseline characteristics compared between anxious vs. non-anxious.

	Anxious		Non-Anxious		*p*
	N	%	N	%	
Male gender	35	79.5	26	78.8	0.94
NYHA class					0.83
I	15	34.1	12	36.4	
II	24	54.5	16	48.5	
III	5	11.4	5	15.2	
Hyperlipidemia	35	79.5	26	78.8	0.94
Hypertension	38	86.4	28	84.8	0.85
Diabetes mellitus	11	25.0	8	24.2	0.97
History of MI	25	56.8	19	57.6	0.82
Smoking	8	18.2	14	42.4	0.02
Beta blocker	39	88.6	30	90.9	0.75
Statins	41	93.2	32	97.0	0.46
ACE-inhibitors	27	61.4	18	54.5	0.55
Antidepressants	12	27.3	6	18.2	0.35
Type D personality	41	77.4	15	62.5	0.01
Socioeconomic status					0.045
Low	15	28.3	10	41.7	
Medium	25	47.2	8	33.3	
High	13	24.5	6	25.0	
	**M**	**SD**	**M**	**SD**	** *p* **
Age, y	57.5	7.8	62.2	8.5	0.01
LVEF	64.5	13.2	62.0	13.6	0.48
BMI	29.5	5.3	29.2	6.0	0.86
CCI	2.3	1.9	2.3	1.3	0.98
Aerobic exercise (min/week)	388.0	254.3	342.3	270.2	0.72
Number of affected coronary vessels	2.3	1.2	2.2	0.8	0.93
Cortisol (nmol/L) +0	9.9	6.1	14.3	8.8	0.01
Cortisol (nmol/L) +30	14.7	6.0	15.0	7.7	0.73
Cortisol (nmol/L) +45	15.7	6.0	13.6	7.5	0.20
Cortisol (nmol/L) +60	12.5	5.8	11.5	5.9	0.97

Abbreviations: BMI = body mass index (kg/m²), CCI = Charlson Comorbidity Index, LVEF = left ventricular ejection fraction, M = mean, MI = myocardial infarction, NYHA = New York Heart Association, SD = standard deviation, *p* = significance level.

## Data Availability

The data presented in this study are available on request from the corresponding author. The data are not publicly available due to the use of confidential data.

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
