# Peer review of "Cortisol Awakening Reaction and Anxiety in Depressed Coronary Artery Disease Patients"

_jcm, 2022, doi:10.3390/jcm11020374_

Round 1
Reviewer 1 Report
The manuscript has been revised appropriately, and I have no further comments.
Author Response
Thank you very much.
Reviewer 2 Report
In their manuscript Weber C et al provide results of a substudy of a moderately large multicenter study, demonstrating the associations between anxiety and cortisol awakening reaction in moderately depressed coronary artery disease patients. The topic is interesting, however there are some major concerns which limit the utility of the results.
- A major limitation of the study is the cortisol sampling. As it was based on the training of patients and was performed in their home by themselves, mistakes could have been made in the circumstances of the sampling and also in the transport of the samples.
- As mean anxiety points were in the border of the definition of anxiety (11.2 vs 11) with small SD, anxiety scale points should also be examined as continuous variable not only as categorical variable. An alternative could be the comparison of tertiles or quartiles based on anxiety points.
- As cortisol level and anxiety are both influenced by stressful events, it would be good to know how long time passed between the heart attack of the patients involved and between the measurements performed.
- About the cardiovascular medications of the patients only beta-blockers are mentioned, statins, ACE-inhibitors or other cardiovascular medications are absent. Also anxiolytic and antidepressant medications.
- Physical health sum score is provided, but as physical activity can also influence cortisol level, anxiety and depression as well, it would be helpful to provide data about the length of aerobic exercise weekly and to consider in regression analysis.
Author Response
Thank you very much for reviewing our manuscript. Please find our answers attached.

Round 2
Reviewer 2 Report
Authors adequately answered my concerns and performed an extensive revision.
This manuscript is a resubmission of an earlier submission. The following is a list of the peer review reports and author responses from that submission.
Round 1
Reviewer 1 Report
The authors investigated the association of cortisol awakening reaction with anxiety in moderately depressed patients with coronary artery disease. They demonstrated that anxious patients showed lower saliva cortisol levels at awakening and at bedtime, but a significantly steeper 30 min increase and higher AUCi after awakening compared to non-anxious patients. These findings are interesting. However, I would like to raise several concerns that should be addressed.
Major comments
- Is there any difference in clinical background between anxious and non-anxious patients? The authors should provide the data table on the clinical back ground of these groups.
- Is there any association of severity of coronary artery disease (i.e. multi-vessel disease, left main disease) with anxiety and cortisol reaction?
- How about the impact of symptom of coronary artery disease on anxiety and cortisol levels?
- Increase in cortisol levels after awakening in the anxious group was higher, but baseline cortisol levels were significantly lower in the anxious group and the absolute cortisol levels at 30-min were comparable. Which is the most critical parameter, absolute levels, increase levels, or AUC?
- Is there any correlation between cholesterol reaction and severity of anxiety (HADS score)?
- Please provide the data table on cortisol reaction in the anxious and non-anxious group.
- The authors should provide data table of the results of correlation analyses.
- They should also provide data table of repeated-measure ANOVA with confounders.
- The authors previously reported the association between cortisol awakening reaction and anxiety in patients with coronary artery disease. Why did the authors do same analysis in moderately depressed patients with coronary artery disease?
Reviewer 2 Report
This manuscript by Weber et al investigates the cortisol response at waking in patients with depression and coronary artery disease (CAD).
The authors present data suggesting that patients with mild to moderate depression and CAD have a unique cortisol response at waking that is dependent upon whether they also have anxiety.
The working hypothesis is that the steeper increase in cortisol at waking in those patients with anxiety is linked to greater CVD overall. The data showed a negative correlation between total cortisol out put with LVEF, and a and trend between anxiety and and steeper slope of cortisol release.
The study takes the interesting approach to test the combination of depression and anxiety in the one study and investigate the impact on cortisol release and total levels. This dissects 2 important physiological aspects of cortisol release and allows for a clearer understanding of the combined effect of anxiety and depression on cortisol and CVD.
The limitations of the study are well described and accounted for. Although a group of patients without depression/anxiety but with CAD would have been an interesting point of comparison to understand the effect on cortisol release. A prospective study using independent subjects would be in the authors sights.